# BAKU: An Efficient Transformer for Multi-Task Policy Learning

**Siddhant Haldar**      **Zhuoran Peng**      **Lerrel Pinto**

New York University

## Abstract

Training generalist agents capable of solving diverse tasks is challenging, often requiring large datasets of expert demonstrations. This is particularly problematic in robotics, where each data point requires physical execution of actions in the real world. Thus, there is a pressing need for architectures that can effectively leverage the available training data. In this work, we present BAKU, a simple transformer architecture that enables efficient learning of multi-task robot policies. BAKU builds upon recent advancements in offline imitation learning and meticulously combines observation trunks, action chunking, multi-sensory observations, and action heads to substantially improve upon prior work. Our experiments on 129 simulated tasks across LIBERO, Meta-World suite, and the Deepmind Control suite exhibit an overall 18% absolute improvement over RT-1 and MT-ACT, with a 36% improvement on the harder LIBERO benchmark. On 30 real-world manipulation tasks, given an average of just 17 demonstrations per task, BAKU achieves a 91% success rate. Videos of the robot are best viewed at baku-robot.github.io.

## 1 Introduction

Learning generalist policies that can solve multiple tasks is a long standing problem in decision making and robotics. While significant advances have been made in computer vision [4, 59] and natural language processing [2, 53, 69], algorithms that can effectively do so for physical agents are far behind. A key reason for this is the scale of available data. While large-scale datasets in vision and language can readily be amassed from the Internet, robotics presents a unique challenge. Given its interactive nature, data acquisition requires physical engagement with the world, making robot data considerably more laborious to obtain in terms of both time and financial costs.

A prominent approach for training multi-task policies is to bite the bullet and collect large amounts of data, often by contracting teleoperators [37, 6, 61]. However, policies trained on such data are quite inefficient, often achieving performance far below independently trained single-task policies [75, 44, 57]. The current best answer to solve this problem is unfortunately to collect even more demonstration data from experts.

In this work, we present BAKU, a simple architecture for multi-task policy learning that provides highly efficient training, particularly in data-scarce problems such as robotics. BAKU builds upon recent work in multitask learning [5, 6] and has three key features. First, a transformer encoder that fuses information from multiple modalities like vision and language while incorporating temporal context. Second, a FiLM-conditioned [46] visual encoder helps the model learn task-specific representations by adapting the encoder to the task. Third, an action prediction head that is separated from the observational encoding trunk, enabling BAKU to be easily retrofitted with state-of-the-art action

---

Correspondence to: siddhanthaldar@nyu.edu

38th Conference on Neural Information Processing Systems (NeurIPS 2024).

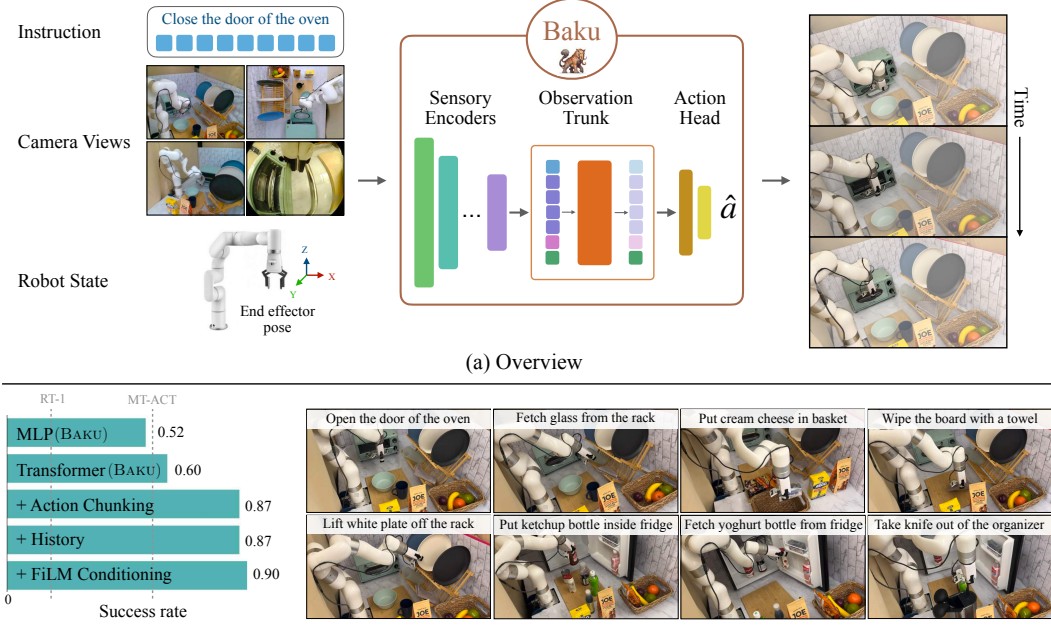

(a) Overview

(b) Performance on LIBERO-90

(c) Real-world tasks (8 of 30 shown)

Figure 1: **(a)** We present BAKU, a simple transformer architecture learning multi-task policies across a diverse range of tasks. BAKU encodes inputs from different modalities using modality-specific encoders. The encoded representations are merged in an observation trunk before predicting actions through an action head. **(b)** We develop a unified policy for 90 tasks in the LIBERO-90 benchmark, discussing design choices that impact multi-task performance. **(c)** On our xArm robot, BAKU can learn a single multi-task policy for 30 tasks with an average of 17 demonstrations collected per task.

generation models [39, 60, 31, 10]. The novelty of BAKU hence lies in carefully combining these ideas to produce a new transformer architecture particularly suited for multitask decision making.

To demonstrate the effectiveness of BAKU, we run extensive experiments on 129 simulated tasks across LIBERO [34], Meta-World [76], and DeepMind Control [67], and 30 robotic manipulation tasks on an xArm robot (see Fig. 1). Our main findings are summarized below:

1. BAKU exhibits an overall 18% absolute performance improvement over prior state-of-the-art multi-task learning algorithms on 129 tasks across 3 simulated environment suites (Section 4.1). BAKU sets a state-of-the-art performance on LIBERO with 90% average success rate, a 36% absolute improvement over prior work (Table 1).

2. On real-world tasks, with an average of 17 demonstrations per task, BAKU achieves an average success rate of 91% across 30 diverse tasks in a multi-task kitchen environment, with randomized object initialization. This outperforms prior state-of-the-art algorithms by 35% (Section 4.2).

3. Through an ablation analysis, we study the importance of each component in BAKU (Section 4.4), particularly the role of action chunking [77] and a multimodal action head in boosting performance, especially in our real-world experiments.

All of our datasets, and training and evaluation code will be made publicly available. Videos of our trained policies can be seen here: baku-robot.github.io.

## 2   Background

**Imitation Learning:**    The goal of imitation learning is to learn a behavior policy $\pi^b$ given access to either the expert policy $\pi^e$ or trajectories derived from the expert policy $\mathcal{T}^e$. While there are a multitude of settings with differing levels of access to the expert [68], this work operates in the setting where the agent only has access to observation-based trajectories, i.e. $\mathcal{T}^e \equiv \{(o_t, a_t)_{t=0}^T\}_{n=0}^N$. Here

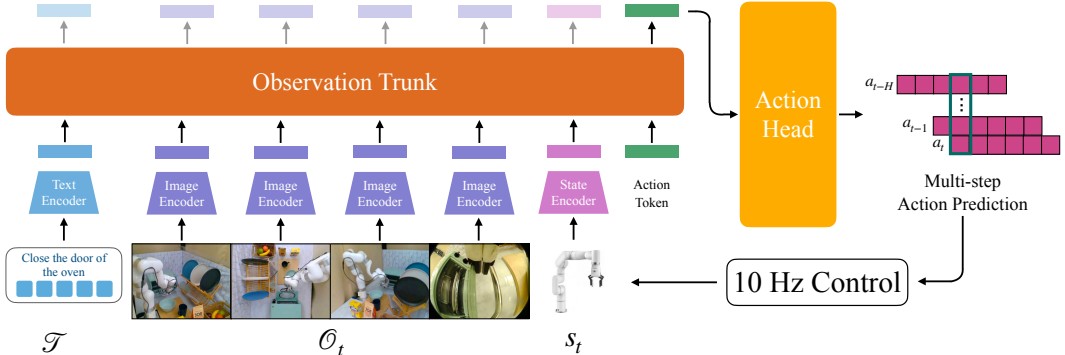

Figure 2: Overview of BAKU, broken down into modality-specific sensory encoders, an observation trunk, and an action head predicting a chunk of actions. BAKU takes as input observations from multiple camera views $\mathcal{O}_t$, robot proprioceptive state $s_t$ and a task instruction $\mathcal{T}$ and enables performing closed-loop control at 10Hz in our real world experiments on the xArm.

$N$ and $T$ denote the number of trajectory rollouts and episode timesteps respectively. We choose this specific setting since obtaining observations and actions from expert or near-expert demonstrators is feasible in real-world settings [77, 24] and falls in line with recent work in this area [77, 11, 31, 10].

**Multi-task Behavior Cloning:** Behavior Cloning (BC) corresponds to solving the maximum likelihood problem shown in Eq. 1. Here $\mathcal{T}^e$ refers to expert demonstrations. When parameterized by a normal distribution with fixed variance, the objective can be framed as a regression problem where, given observations $o^e$, $\pi^{BC}$ needs to output $a^e$.

$$\mathcal{L}^{BC} = \mathbb{E}_{(o^e, a^e) \sim \mathcal{T}^e} \|a^e - \pi^{BC}(o^e)\|^2 \tag{1}$$

After training, it enables $\pi^{BC}$ to mimic the actions corresponding to the observations seen in the demonstrations. In multi-task settings, we use the same formulation for BC but condition the action prediction on a goal variable $g^e$. Thus, the loss function for multi-task BC becomes the following.

$$\mathcal{L}^{BC} = \mathbb{E}_{(o^e, a^e, g^e) \sim \mathcal{T}^e} \|a^e - \pi^{BC}(o^e|g^e)\|^2 \tag{2}$$

In this work, we represent goals as either a text description of the task [6, 5] or a goal image [11, 72].

## 3 BAKU

The design of multi-task learning algorithms involves numerous decisions regarding model architecture and component selection. This often results in complex architectures where the importance of individual components is sometimes unclear. In this work, we perform a systematic and thorough ablation study across the various multi-task learning architectures proposed by prior works [6, 5, 63] and introduce BAKU, a simple architecture for multi-task policy learning. To facilitate our analysis, we divide the overall model architecture into three main components: sensory encoders, an observation trunk, and an action head. Sensory encoders process raw sensor inputs from different modalities into useful feature representations. The observation trunk combines the encoded information from the different modalities. Finally, the action head utilizes the combined information to predict actions. Below, we describe these three components in detail, with additional algorithmic details provided in Appendix A.

### 3.1 Sensory Encoders

In the real-world, robots encounter diverse data modalities, including vision, depth feedback, proprioceptive feedback, and task instructions in various forms such as text, goal images, or task videos. In BAKU, we focus on vision, robot proprioception, and text or goal image based task instructions. For vision, we use a ResNet-18 [19] visual encoder to process images of the scene, enhanced with

a FiLM [46] layer to integrate task-specific information. Robot proprioception data is processed through a two-layer multilayer perception (MLP) encoder. For text, we use a 6-layer version of MiniLM [73] provided in Sentence Transformers [54]. We project the representations obtained from all modalities to the same dimensionality through additional MLP layers, to facilitate combining the encoded information. We have included a description of FiLM conditioning in Appendix A.1.

### 3.2 Observation Trunk

The encoded inputs from all sensory modalities are combined in the observation trunk. We explore two variants of the trunk network:

**Multilayer Perceptron (MLP)** The encoded inputs are concatenated into a single feature vector and passed through a multilayer perceptron. When using a history of observations, the inputs corresponding to all time steps are concatenated.

**Transformer** Each encoded input is treated as an observation token and passed through a transformer decoder network [70]. A learnable action token is appended to the list of observation tokens and used to predict the action. When using a historical observation, a separate action token is added for each time step to enable predicting actions for all time steps. A causal mask is applied to the transformer to ensure that actions are predicted solely based on past observations.

Both variants output action feature vectors (corresponding to the action tokens for a transformer), which are then passed through an action head to predict actions.

### 3.3 Action Head

The final component of our architecture is the action head, an action prediction module that takes as input the action feature vectors obtained from the observation trunk and predicts the corresponding actions. An independent action prediction module enables us to unify several state-of-the-art action generation models within the same framework. We experiment with five action head variants: vanilla MLP, Gaussian Mixture Model (GMM) [36], Behavior Transformer (BeT) [60], Vector-Quantized Behavior Transformer (VQ-BeT) [31], and diffusion policy [45, 10, 55]. Considering the temporal correlation in robot movements, we follow prior work [77, 5] and include action chunking with exponential temporal averaging to produce smoother behaviors and counteract the covariate shift often seen in low-data imitation learning scenarios. In contrast to previous works [77, 5] that decode actions for each time step separately, we predict the action chunk as a single concatenated vector. We find that this simplification improves performance (see Table 1). More details about each action head variant has been provided in Appendix A.2 along with details about the exponential temporal smoothing technique in Appendix A.3.

### 3.4 Putting it all together

Our proposed architecture is depicted in Figure 2. Through extensive experimentation (see Section 4), our final architecture includes a modified FiLM-conditioned ResNet-18 vision encoder (provided with the LIBERO benchmark [34]), an MLP encoder for robot proprioception, and a pre-trained text encoder for task instructions. For environments with multiple camera views, we use a common visual encoder across all views. We provide only the current observation as an input and the observation trunk uses a causal transformer decoder architecture [29]. The base version of our model uses an MLP action head with action chunking and temporal smoothing to produce smoother motions. We also experiment with multimodal variants of action heads and have provided the results in Section 4.4.

The parameter counts are approximately 2.1M for the sensory encoders, 6.5M for the observation trunk, and 1.4M for the action head, bringing the total model size to approximately 10M parameters.

## 4   Experiments

Our experiments are designed to answer the following questions: (a) How well does BAKU work for multi-task learning? (b) How does BAKU perform on real-world tasks? (c) How does BAKU perform

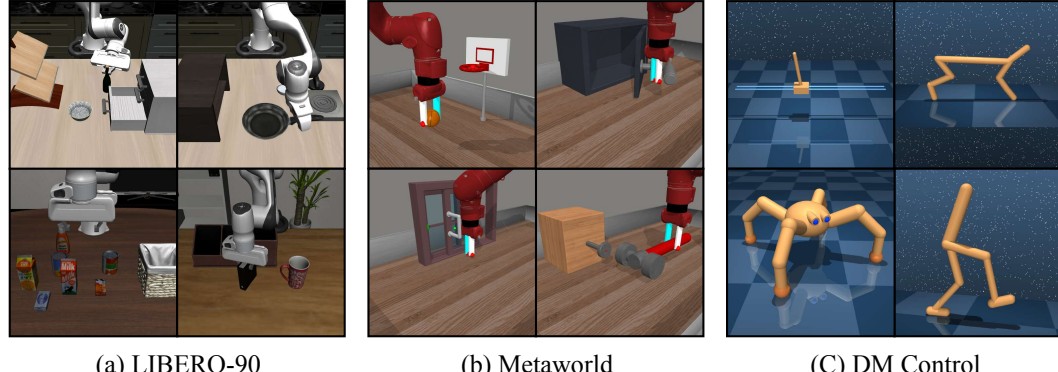

| (a) LIBERO-90 | (b) Metaworld | (C) DM Control |

Figure 3: BAKU is evaluated on 3 simulated benchmarks - LIBERO, Meta-World, and DM Control.

on long-horizon tasks? (d) What design decisions affect multi-task policy learning? Additional results and analysis have been provided in Appendix D.

**Simulation Tasks:** We experiment with 90 manipulation tasks from the LIBERO-90 benchmark [34], 30 manipulation tasks from Meta-World suite [76], and 9 locomotion tasks from DeepMind Control Suite (DMC) [67]. Figure 3 depicts the simulated environments. For LIBERO-90, we use 50 demonstrations per task provided with the benchmark and use images from third-person and gripper camera views, as well as robot proprioception as input. For Meta-World, we obtain 35 demonstrations per task from an expert policy trained with demonstration-guided reinforcement learning [17, 18], using only the third-person view as input. We use images of size $128 \times 128$ for LIBERO-90 and $84 \times 84$ for Meta-World. For DMC, we train state-based locomotion policies using 500 demonstrations per task obtained from experts trained with DrQ-v2 [74]. All evaluations are conducted using 10 policy rollouts per task. More details about the simulated environments can be found in Appendix B.

**Robot Tasks:** Our real-world experiments are performed on a Ufactory xArm 7 robot with an xArm Gripper in a multi-task kitchen environment. The policies are trained on RGB images of size $128 \times 128$ obtained from four different camera views, including an egocentric camera attached to the robot gripper. The action space comprises the robot end effector pose and the gripper state. We collect a total of 520 demonstrations across 30 tasks, averaging 17 demonstrations per task. The demonstrations were collected using a VR-based teleoperation system [24] at a 30Hz frequency. The learned policies are deployed at 10Hz. Figure 4 shows selected tasks from our real-world environment. More details about each task and robot control can be found in Appendix C.

**Strong Baselines:** In this section, we provide a detailed explanation of our baselines in relation to BAKU.

1. **MT-ACT [5]:** Multi-task Action-Chunking Transformer (MT-ACT) is a state-of-the-art transformer encoder-decoder architecture for learning multi-task policies. MT-ACT extends Action-Chunking Transformer (ACT) [77] to a multi-task setting. MT-ACT takes as input observations from multiple camera views, robot proprioception, and task instructions. Each input modality passes through dedicated encoders. The encoded observations are then fused in a transformer encoder, the output of which conditions a transformer decoder to predict chunks of future actions. Each predicted action corresponds to a position embedding input to the decoder. In ACT and MT-ACT, a conditional variational autoencoder (CVAE) is used to learn a multimodal style variable which conditions the encoder to deal with multimodal action distributions. During inference, the style variable is set to zero, leading to unimodal behavior. In contrast, BAKU uses a decoder-only transformer architecture that directly predicts action features corresponding to past observations. This enables us to (1) leverage recent advances in multimodal action generation by plugging in several unimodal and multimodal heads for action prediction, and (2) incorporate a history of observations to predict actions for each time step in the history (see Section 4.4 for results on the

Table 1: Performance of multi-task policies learned using BAKU on 3 simulated benchmarks - LIBERO-90, Meta-World, and DM Control - and a real xArm robot. We observe that BAKU significantly outperforms prior work on both simulated and real world tasks.

| Method | LIBERO-90 (90 tasks) | Meta-World (30 tasks) | DMC (9 tasks) | Real Robot (20 tasks) |
|---|---|---|---|---|
| RT-1 | 0.16 | 0.65 | 0.66 | 0.37 |
| MTACT | 0.54 | 0.13 | 0.59 | 0.56 |
| BAKU (**Ours**) | **0.9** | **0.79** | **0.7** | 0.86 |
| BAKU w/ VQ-BeT (**Ours**) | **0.9** | 0.78 | **0.7** | **0.91** |

use of observation history). Further, using a multimodal action head enables BAKU to exhibit multimodal behavior during inference, improving real-world performance (Table 1).

2. **RT-1 [6]:** RT-1 is a transformer-based multi-task policy learning architecture that models actions as discrete classes by uniformly discretizing them into bins. RT-1 uses a FiLM-conditioned vision encoder (ResNet-18 in our implementation), but instead of directly using the final 512-dimensional representation, it splits an intermediate feature map of size $k \times k \times 512$ into $k^2$ tokens of 512 dimensions each. These tokens are passed through a Token Learner [58] module to reduce them to 8 tokens per image. The reduced number of tokens is then passed through a decoder-only transformer architecture to predict a discrete action. In contrast, BAKU directly uses the final 512-dimensional representation from the vision encoder, without summarizing tokens via a token learner. Additionally, BAKU predicts a continuous action through an unimodal or multimodal action head. Based on our experiments (Table 1), we observe that these design choices in BAKU lead to significant improvements in performance over RT-1.

### 4.1 How well does BAKU work for multi-task learning?

We evaluate the multi-task performance of BAKU on 90 tasks from the LIBERO-90 benchmark, 30 tasks from Meta-World, and 9 tasks from DMC. Table 1 compares the performance of BAKU with our baselines, RT-1 [6] and MT-ACT [5]. BAKU outperforms the strongest baseline by 36% and 14% on LIBERO-90 and Meta-World respectively, demonstrating more effective multi-task learning on complex manipulation tasks. On the simpler DMC locomotion tasks, BAKU outperforms the strongest baseline by 4%. Overall, these results suggest that BAKU more effectively leverages relationships between tasks to achieve superior multi-task learning performance compared to prior methods.

### 4.2 How does BAKU perform on real-world tasks?

We evaluate BAKU on 30 manipulation tasks in our real-world kitchen environment, comparing it with MT-ACT and RT-1. During evaluations, the xArm was always initialized at the same pose and the objects being manipulated were placed in a fixed set of positions for all methods. We conducted 5 evaluation runs per task, totaling 150 evaluation runs per method. Table 1 includes our real-world results. We observe that BAKU achieves an 86% success rate across all tasks, outperforming the strongest baseline by 30%. Replacing the MLP action head with a multimodal VQ-BeT [31] head further improves the success rate to 91%, outperforming the strongest baseline by 35%. Figure 4 shows real-world rollout trajectories for a selected task set with all tasks included in Appendix C. Appendix D.1 provides the task-wise performance for each method. Overall, these results indicate BAKU's promise for deploying multi-task policies on real-world robotic systems.

### 4.3 How does BAKU perform on long-horizon tasks?

We also evaluate BAKU on long-horizon tasks in the simulated LIBERO-10 benchmark and our real-world multi-task kitchen environment. Table 2 provides the results on 10 tasks in LIBERO-10 and 5 long-horizon tasks in the real kitchen environment, each composed of two shorter tasks chained sequentially. We use 50 demonstrations per task for LIBERO-10 and an average of 19 demonstrations per task for the real robot. We observe that BAKU significantly outperforms our strongest baseline, MT-ACT, on these long horizon tasks, achieving on average 19% higher success rate. This highlights BAKU's ability to learn policies that can effectively plan and execute sequences of actions over

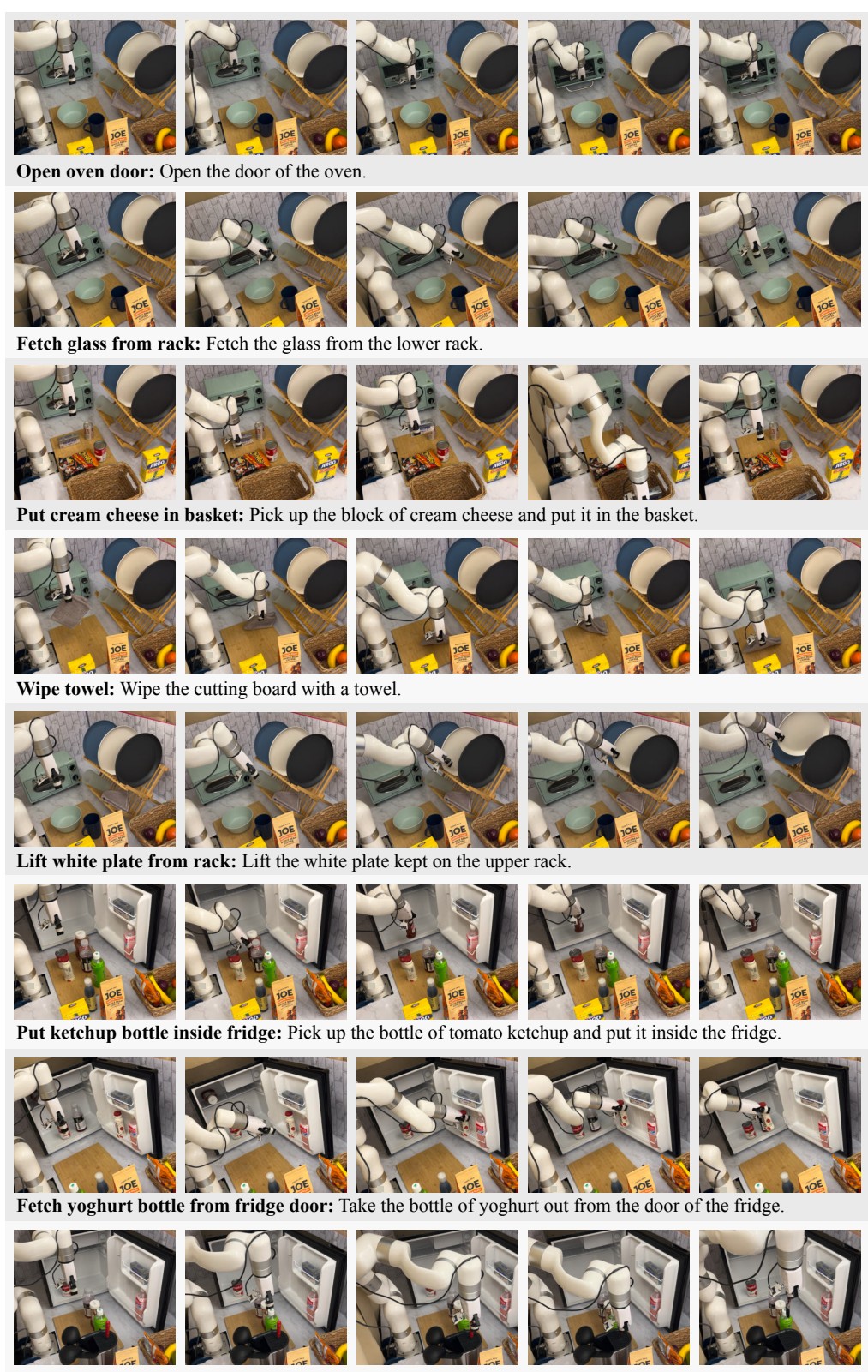

Figure 4: Real-world policy rollouts showing BAKU's capability in complex manipulation tasks.

Table 2: Performance of multi-task policies learned using BAKU on long-horizon tasks in the LIBERO-10 simulated benchmark and a real xArm robot. We observe that BAKU significantly outperforms prior work on both simulated and real world tasks.

| Method | LIBERO-10 (10 tasks) | Real Robot (5 tasks) |
|---|---|---|
| MT-ACT | 0.68 | 0.64 |
| BAKU **(Ours)** | **0.86** | **0.84** |

extended time horizons. Real world rollouts of these long-horizon tasks have been included in Appendix C with the task-wise performance and demonstration details in Appendix D.1.

## 4.4 What design decisions affect multi-task policy learning?

As described in Section 3, our multi-task policy architecture consists of three main components: sensory encoders, an observation trunk, and an action head. In this section, we analyze the design choices within each component and their effect on overall multi-task performance. We consider BAKU with an MLP action head (described in Section 3.4) as our base model. For ablations, we vary only a single property at a time while keeping all other aspects identical. This experimental setup allows us to clearly isolate the impact of individual design decisions. We examine different observation trunks, model sizes, action heads, goal modalities, and the use of action chunking, observation history, and task conditioning through FiLM [46]. The results of our ablation study are provided in Table 3 with more analysis in Appendix D. The results provide insights into which components and properties are most important for effective multi-task learning with BAKU.

**Effect of Observation Trunk:** We experiment with two trunk types: an MLP and a transformer architecture. In Table 3, we observe a slight performance dip when using an MLP trunk on Meta-World and DMC. For LIBERO-90, our most complex simulated benchmark, an MLP trunk resulted in a 9% lower success rate than a transformer trunk. This highlights the efficacy of transformers for modeling complex relationships between observations from multiple sensing modalities and actions.

**Effect of Model Size:** We study the effect of model size on multi-task performance by evaluating configurations with 4.4M, 10M, 31M, and 114M parameters. For each variant, we vary the size of the observation trunk and the action head while keeping the sensory encoders constant. The results in Table 3 show that the 4.4M, 10M, and 31M parameter models achieve similar performance across benchmarks. Surprisingly, the largest 114M parameter model severely underperforms on the harder LIBERO-90 benchmark. We suspect this poor performance may have been due to overfitting on the training data with a larger capacity. Based on these results, we use the 10M parameter model for BAKU since it is the smallest model with the best performance on 2 of the 3 simulated benchmarks.

**Effect of Action Head:** We compare the performance of BAKU retrofitted with five different action heads: MLP, GMM [36], BeT [60], VQ-BeT [31], and diffusion [10]. Having an independent action head enables us to extend these state-of-the-art action generation models to multi-task settings. Table 3 shows that on simulated benchmarks, a simple MLP head performs just as well or better than other multimodal action heads. Among the multimodal variants, VQ-BeT achieves the best performance. As a result, we also evaluate BAKU with a VQ-BeT action head in our real-world setup (Table 1). On the real robot, having a multimodal action head proves advantageous with the VQ-BeT head achieving a success rate of 91%, a 5% improvement over an MLP action head. Hence, our experiments demonstrate that while multimodal heads may provide benefits for real-world deployment, a simple MLP head can perform well on simulated data with limited behavioral diversity.

**Effect of Action Chunking:** We study the effect of action chunking on multi-task policy performance. For the image-based LIBERO-90 and Meta-World benchmarks, we predict a chunk of 10 future actions. For the locomotion tasks in DMC, we predicted 3 future actions. Based on the results in Table 3, we observe the largest difference on LIBERO-90, where removing action chunking and instead predicting a single action led to a 14% drop in performance. In contrast, there is no perceptible difference in performance on Meta-World with and without chunking. For the locomotion domains in

Table 3: Study of design decisions for BAKU that affects multi-task performance.

| Category | Variant | LIBERO-90 | Meta-World | DMC |
|---|---|---|---|---|
| Observation Trunk | MLP | 0.81 | 0.78 | 0.68 |
| | Transformer | **0.90** | **0.79** | **0.70** |
| Model Size | 4.4M | 0.85 | 0.78 | 0.68 |
| | 10M | **0.9** | 0.79 | **0.70** |
| | 31M | 0.87 | **0.81** | **0.70** |
| | 114M | 0.19 | **0.81** | 0.68 |
| Action Head | MLP | **0.90** | **0.79** | **0.70** |
| | GMM | 0.84 | 0.65 | 0.67 |
| | BeT | 0.89 | 0.78 | 0.60 |
| | VQ-BeT | **0.90** | 0.78 | **0.70** |
| | Diffusion | 0.89 | 0.45 | 0.61 |
| Action Chunking | ✗ | 0.76 | 0.78 | **0.74** |
| | ✓ | **0.90** | **0.79** | 0.70 |
| Observation History | ✗ | **0.90** | 0.79 | **0.70** |
| | With last-step loss | 0.54 | 0.08 | 0.37 |
| | With multi-step loss | **0.90** | **0.82** | 0.68 |
| Goal Modality | Text | 0.90 | 0.79 | N/A |
| | Goal Image | 0.88 | **0.81** | N/A |
| | Intermediate Image | **0.91** | 0.80 | N/A |
| FiLM | ✗ | 0.87 | **0.79** | N/A |
| | ✓ | **0.90** | **0.79** | N/A |

DMC, we see a 4% performance increase when removing chunking. Hence, action chunking benefits manipulation tasks while mildly hindering locomotion tasks from our experiments.

**Effect of Observation History:** We study the effect of using an observation history on multi-task performance. As shown in Table 3, naively using an observation history where the action prediction loss is only computed for the last time step significantly degrades performance. However, since BAKU uses a transformer observation encoder, it allows predicting actions for all observations in the history and computing the prediction loss over all time steps. Empirically, we found this multi-step prediction loss provides richer supervision and improves the single-step loss performance by an average of 47% across all benchmarks. However, incorporating an observation history with multi-step action prediction did not noticeably improve overall policy performance compared to using no history. Therefore, our final architecture only uses the most recent observation as an input.

**Effect of Goal Modality:** We experiment with 3 different goal modalities: text instruction, goal image, and intermediate goal image. The text instructions are directly obtained from the task data. The goal image is obtained by randomly sampling a demonstration from the training dataset and taking the last frame. For an intermediate goal image, we consider this randomly sampled task demonstration, and for every time step, treat the frame $k$ steps in the future as the goal image [72]. Table 3 contains the results on LIBERO-90 and Meta-World, as goal images do not apply to the state-based DMC tasks. We set $k$ to 50 steps for LIBERO-90 and 30 steps for Meta-World. Since LIBERO-90 has two camera views, we use the third-person view to obtain goal images. We observe that all three goal modalities show a similar performance with slight variations. Overall, our approach supports different goal representations with only minor variations in performance.

**Effect of FiLM Conditioning:** We examine the impact of using a FiLM-conditioned vision encoder for language-guided multi-task policies. As shown in Table 3, on the image-based LIBERO-90 and Meta-World benchmarks, a FiLM-conditioned vision encoder performs equally well or better than an unconditional encoder. FiLM conditioning allows modulating the vision encoder's parameters conditioned on the language input. This provides an effective way to fuse visual and linguistic information for solving tasks. Therefore, BAKU employs a FiLM-conditioned vision encoder for our image-based experiments.

## 5 Related Work

**Imitation Learning (IL)**  IL [23] refers to the setting where agents learn from demonstrations without access to environment rewards. IL can be broadly categorized into Behavior Cloning (BC) [48, 68] and Inverse Reinforcement Learning (IRL) [41, 1]. BC solely learns from offline demonstrations but suffers on out-of-distributions samples [56] whereas IRL focuses on learning a robust reward function through online interactions but suffers from sample inefficiency [17, 18]. In this work, we focus on using BC for learning multi-task policies. In recent years, there have been significant advances in single-task behavior cloning with the development of multimodal action generation models using GMMs [36, 39], EBMs [13], BeT [60, 11, 31], and diffusion [45, 10, 55, 7]. There has also been notable progress in solving long-horizon tasks through imitation learning with some works relying solely on robot data [38, 21, 9, 77, 33, 65] while others attempt to bootstrap learning from human data [72]. Further, these advances in policy learning combined with significant strides in self-supervised representation learning [8, 42, 20] have enabled deploying these policies in messy and unpredictable environments such as our homes [61] as well as zero-shot deployment in-the-wild [62, 64]. However, despite the large body of work advancing single-task robotic policy learning, there still exists a gap between single-task and multi-task performance for policy learning [75, 44, 57].

**Multi-task Learning**  Robotics has a long history of multi-task learning. There is a significant body of work focusing on learning policies for robotic grasping with the aim of generalizing to new tasks [32, 47, 14, 71, 12], robotic language understanding [40, 35, 66, 3], and framing multi-task learning as a goal reaching problem [51, 28, 22]. Additionally, several works have collected varied multi-task robotics datasets [38, 34, 5, 43, 30]. Recently, there has been an increased use of transformer-based architectures for multi-task robot learning, spanning across robot navigation [62, 64], locomotion [27, 15, 49, 50], and manipulation [6, 78, 11, 5]. While most of these works use text conditioning for task specification, some go beyond text to use goal images [11, 16] and videos [26, 25] as well. Another emerging trend is co-training these robot policies with additional tasks, such as visual question answering and image captioning [52, 78], to develop more generalizable policies. Overall, multi-task learning has been widely applied in robotics and, more recently, using high-capacity transformer models to learn robot control policies has become common practice in the field. Despite their effectiveness, the architectures for these policies often become complicated, with the necessary components sometimes being unclear. Our proposed model, BAKU, combines key ideas from prior work into a single architecture to produce a model that is both simple and outperforms state-of-the-art methods in multi-task policy learning.

## 6 Conclusion and Limitations

In this work, we presented BAKU, a simple transformer architecture that demonstrates improved multi-task policy learning performance on a variety of simulated and real-world domains compared to prior state-of-the-art methods. We recognize a few limitations in this work: (a) In our real-world experiments, while BAKU achieved good performance on most tasks, it struggled on some precise manipulation tasks, such as *opening an oven door* or *placing a tea bottle in the fridge*. This suggests that data sharing across tasks of varying difficulty may hinder performance on more precise skills. Developing techniques to learn a single policy for different task complexity levels could help address this. (b) Currently, we focus on performing a single skill at a time. Developing algorithms capable of chaining multiple such skills can enable effective long-horizon robot manipulation. (c) In this work, we primarily studied the policy architecture and did not analyze the generalization benefits of multi-task learning as the number of tasks increases. A study of the emergence of such generalization with greater task diversity would be another interesting direction. Overall, we hope that BAKU serves as an important step towards developing multi-task policies capable of performing precise robotic manipulation.

## Acknowledgments and Disclosure of Funding

We thank Nur Muhammad Shafiullah, Ulyana Piterbarg, Ademi Adeniji, Ben Evans, Gaoyue Zhou, and Irmak Güzey for valuable feedback and discussions. This work was supported by grants from Honda, Google, NSF award 2339096 and ONR awards N00014-21-1-2758 and N00014-22-1-2773. LP is supported by the Packard Fellowship.

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

# A  Algorithmic Details

## A.1  FiLM Conditioning

Feature-wise Linear Modulation (FiLM) [46] is a technique used for conditioning neural networks that allows the network to modulate its behavior based on an external conditioning signal, such as text instructions or observations. In the context of text conditioning for policy learning, the text instructions are first encoded into a conditioning vector. This conditioning vector is then used to modulate the activations of the neural network through FiLM layers. FiLM applies a feature-wise affine transformation (scaling and shifting) to the activations of the network, conditioned on the text embedding. In other word, assuming $\mathbf{x}$ is a FiLM layer's input, $\mathbf{z}$ is a conditioning input, and $\gamma$ and $\beta$ are $\mathbf{z}$-dependent scaling and shifting vectors,

$$FiLM(\mathbf{x}) = \gamma(\mathbf{z}) \odot \mathbf{x} + \beta(\mathbf{z}) \tag{3}$$

This allows the network to adapt its computation and output based on the given text instructions, enabling tasks like instruction following or conditioning the policy on language descriptions.

## A.2  Action Heads

Having a separate action prediction module allows BAKU to leverage state-of-the-art techniques for action generation. In this work, we evaluate five different action head variants. Below we briefly describe each variant. For more details on these methods, please refer to the original publications.

**Multilayer Perceptron (MLP)** This is a simple neural network comprising multiple dense layers. We use a two-layer MLP for our experiments.

**Gaussian Mixture Model (GMM) [36]** A Gaussian mixture model (GMM) action head models the policy as a mixture of Gaussians, enabling multi-modal action sampling for continuous control problems. The GMM parameters are part of the learned policy network. For our experiments, we employ a two-layer GMM action head with five action modes and a Softplus activation function.

**Behavior Transformer (BeT) [60]** The Behavior Transformer (BeT) models continuous action prediction as a two-part problem. Actions in the training data are first clustered into $k$ bins using k-means clustering. A discrete action head classifies the cluster an action belongs to, while an offset action head predicts an offset value added to the corresponding cluster center. The discrete head uses a focal loss, while the offset head uses L2 loss. For our experiments, we use BeT with 64 action clusters.

**Vector-Quantized Behavior Transformer (VQ-BeT) [31]** The Vector-Quantized Behavior Transformer (VQ-BeT) extends BeT by replacing k-means clustering with residual VQVAE-based tokenization, significantly improving performance over BeT. For our experiments, we employ VQ-BeT with two residual VQ layers of codebook size and latent dimension 16 and 256, respectively.

**Diffusion [45, 10, 55]** A diffusion action head models action prediction as a diffusion process that generates actions over time by iteratively denoising samples from a Gaussian distribution. While highly effective for multi-modal distributions, the iterative denoising during inference slows deployment speed. In this work, we use a transformer-based diffusion head introduced by prior work [45, 10]. We use a two-layer diffusion head for our experiments.

## A.3  Temporal smoothing over action chunking

A naïve implementation of action chunking, where a new environment observation in incorporated every $k$ steps can be suboptimal and can result in jerky robot motion. To improve the smoothness in robot motion, we incorporate an exponential temporal ensembling technique, following prior work [77, 5]. Instead of querying the policy every $k$ steps, we query it at every timestep. This results in an overlap in predicted action chunks and at any given timestep, there will be more than one predicted actions. Instead of using only the current action prediction, we use a temporal ensemble to

combine all the past predictions. This temporal ensemble performs a weighted average over these predictions with an exponential weighing scheme $w_i = exp(-m * i)$, where $w_0$ is the weight for the oldest action. The speed for incorporating a new observation is governed by $m$, where a smaller $m$ means faster incorporation. It must be noted that this ensembling incurs no additional training cost, only extra inference-time computation. In our experiments, similar to prior work [77, 5], we find both action chunking and temporal ensembling to be important for producing precise and smooth motion.

### A.4 Hyperparameters

The complete list of hyperparameters is provided in Table 4. For RT-1 [6], we use our implementation with an RT-1 action head that discretizes the continuous action into discrete bins uniformly. For MT-ACT [5], we use the open-source implementation with the default hyperparameters. We vary the action chunk length for MT-ACT for different benchmarks, the values for which have been provided in Table 4.

**Training time**  Below we provide details about the time required to train BAKU on a single NVIDIA RTX A4000 GPU.

1. **LIBERO:** Training for $600k$ steps with a batch size of 64 and 2 camera views and robot proprioception as input requires around 10.5 hours.

2. **Meta-World:** Training for $600k$ steps with a batch size of 64 and 1 camera view as input requires around 8 hours.

3. **DM Control:** Training for $2M$ steps with a batch size of 128 and robot state as input requires around 26 hours.

4. **xArm Robot:** Training for $200k$ steps with a batch size of 64 and 4 camera views and robot proprioception as input requires around 6 hours.

## B  Simulation Tasks

We evaluate BAKU on three simulated benchmarks: LIBERO-90 [34], MetaWorld [76], and DM Control [67]. For LIBERO-90, we directly use the dataset provided, which includes demonstrations for all 90 tasks. For details on the specific LIBERO-90 tasks, please refer to the original paper [34]. For MetaWorld and DM Control, we collected demonstrations from expert agents trained with reinforcement learning (RL). We include only the tasks for which we were able to obtain expert demonstration data. Table 5 lists the 30 MetaWorld tasks and 9 DM Control tasks used in our experiments.

## C  Robot Tasks

We evaluate BAKU on 30 tasks in our real-world multi-task kitchen environment. We provide the task description along with policy deployment rollouts with BAKU for each task in Figures 5, 6, 7, 8, and 9. The long-horizon task rollouts have been shown in Figure 10.

**Robot control**  We deploy our learned policies at 10Hz using a high-level controller. To facilitate smooth motion on the robot, we deploy a low-level Minimum-Jerk Controller at 100Hz.

## D  Additional Results and Analysis

### D.1  Real-World Task-wise Results

Table 6 provides the task-wise performance for all 30 tasks in our real-world multi-task kitchen environment. We collect an average of 17 demonstrations per task, with a total of 520 demonstrations across all tasks. Task-wise performance for the real-world long-horizon tasks has been included in Table 7.

Table 4: List of hyperparameters.

| Method | Parameter | Value |
|---|---|---|
| Common | Learning rate | $1e^{-4}$ |
| | Image size | $128 \times 128$ (LIBERO-90, xArm) |
| | | $84 \times 84$ (Meta-World) |
| | Mini-batch size | 64 (LIBERO-90, Meta-World, xArm) |
| | | 128 (DM Control) |
| | Optimizer | Adam |
| | Number of training steps | 600000 (LIBERO-90, Meta-World) |
| | | 2000000 (DM Control) |
| | | 200000 (xArm) |
| | Number of demonstrations | 50 (LIBERO-90) |
| | | 35 (Meta-World) |
| | | 500 (DM Control) |
| | | 15 (xArm) |
| | Transformer architecture | minGPT [29] (with 8 layers and 4 heads) |
| | Action chunk length | 10 (LIBERO-90, Meta-World) |
| | | 3 (DMC) |
| | | 20 (xArm) |
| BAKU | Observation trunk | Transformer |
| | Action head | MLP (base) |
| | | GMM, BeT, VQ-BeT, Diffusion (variants) |
| | Hidden dim | 256 |
| | Observation history | False |
| | Action chunking | True |
| | Intermediate goal steps ($k$) | 50 (LIBERO-90) |
| | | 30 (Meta-World) |
| RT-1 | Observation trunk | Transformer |
| | Action head | MLP (base) |
| | Hidden dim | 512 |
| | Observation history | True |
| | History length | 6 |
| | Action chunking | False |
| MT-ACT | Observation history | False |
| | Action chunking | True |

Table 5: List of tasks in Meta-World and DM Control.

| Meta-World | DM Control |
|---|---|
| basketball-v2 | cartpole swingup |
| bin-picking-v2 | cheetah run |
| button-press-v2 | hopper stand |
| button-press-topdown-v2 | quadruped run |
| button-press-topdown-wall-v2 | quadruped walk |
| button-press-wall-v2 | teacher easy |
| coffee-button-v2 | walker stand |
| coffee-pull-v2 | walker walk |
| coffee-push-v2 | walker run |
| dial-turn-v2 | |
| disassemble-v2 | |
| door-lock-v2 | |
| door-open-v2 | |
| door-unlock-v2 | |
| drawer-close-v2 | |
| drawer-open-v2 | |
| faucet-close-v2 | |
| faucet-open-v2 | |
| hammer-v2 | |
| handle-press-v2 | |
| handle-press-side-v2 | |
| handle-pull-v2 | |
| handle-pull-side-v2 | |
| peg-insert-side-v2 | |
| peg-unplug-side-v2 | |
| plate-slide-v2 | |
| plate-slide-back-v2 | |
| plate-slide-back-side-v2 | |
| plate-slide-side-v2 | |
| shelf-place-v2 | |
| soccer-v2 | |
| stick-push-v2 | |
| sweep-v2 | |
| sweep-into-v2 | |
| window-close-v2 | |
| window-open-v2 | |

## D.2 Additional Analysis

In addition to the analysis in Section 4.4, we provide further comparisons here to better justify our design choices.

**Separate vs. Shared Vision Encoders**  On the LIBERO-90 benchmark, environment observations include images from two camera views. Table 12 compares multi-task performance using either a common encoder for both views or separate view-specific encoders. While separate encoders provide a 2% boost in performance, this minor gain comes at the cost of a 15% increase in parameter count per camera view added (since the visual encoders comprise 1.5M parameters in our 10M parameter model). For our real-world experiments involving 4 camera views, this parameter increase would be even more significant. Therefore, in BAKU, we use a shared encoder for all views to keep the model compact, assisting with faster inference speeds.

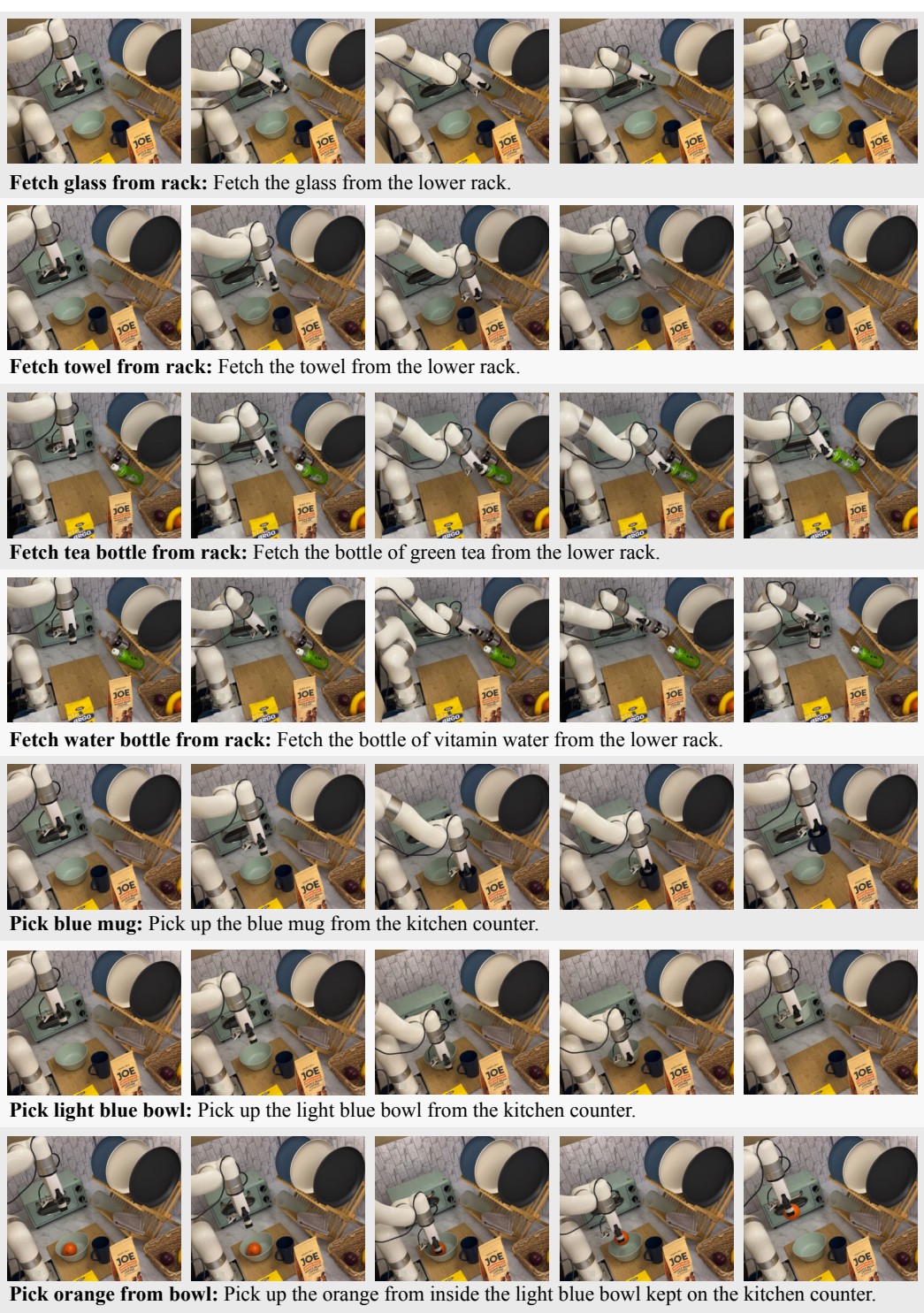

Figure 5: Real-world policy rollouts showing BAKU's capability in complex manipulation tasks.

**Data Efficiency Analysis** We analyze the performance of BAKU with varying number of demonstrations in Table 8 and Table 9. We observe that at each level of data availability, BAKU shows a significantly higher success rate than MT-ACT and RT-1.

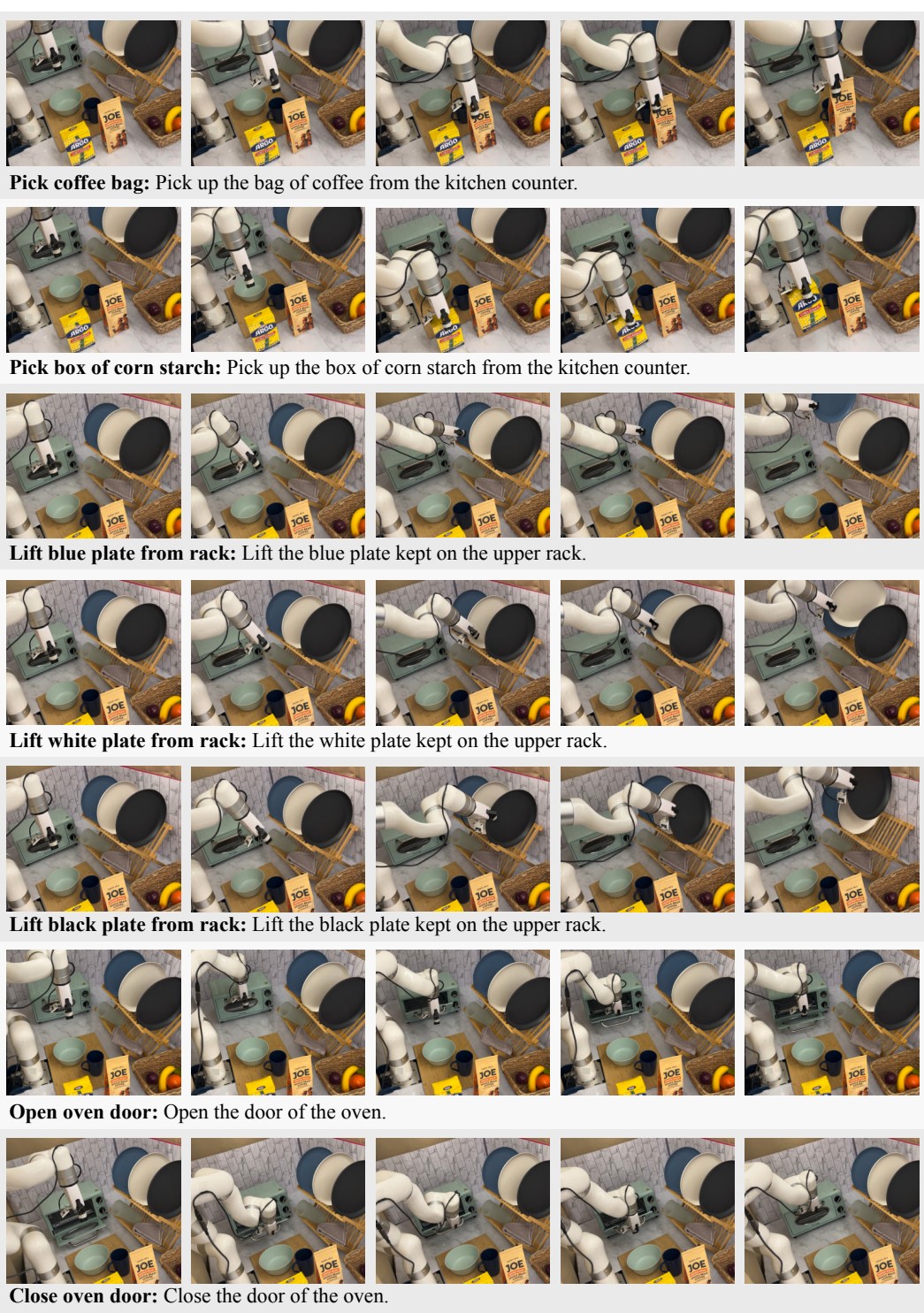

**Pick coffee bag:** Pick up the bag of coffee from the kitchen counter.

**Pick box of corn starch:** Pick up the box of corn starch from the kitchen counter.

**Lift blue plate from rack:** Lift the blue plate kept on the upper rack.

**Lift white plate from rack:** Lift the white plate kept on the upper rack.

**Lift black plate from rack:** Lift the black plate kept on the upper rack.

**Open oven door:** Open the door of the oven.

**Close oven door:** Close the door of the oven.

Figure 6: Real-world policy rollouts showing BAKU's capability in complex manipulation tasks.

**Robustness to training seeds** We provide results on BAKU, RT-1, and MT-ACT across 3 seeds in Table 10. We observe that all three methods are robust to different seed values. Further, probabilistic approaches like GMM and diffusion might be sensitive to favorable seed values, and evaluating on a single seed might make the result unreliable. Thus, Table 11 includes results across 3 seeds on BAKU

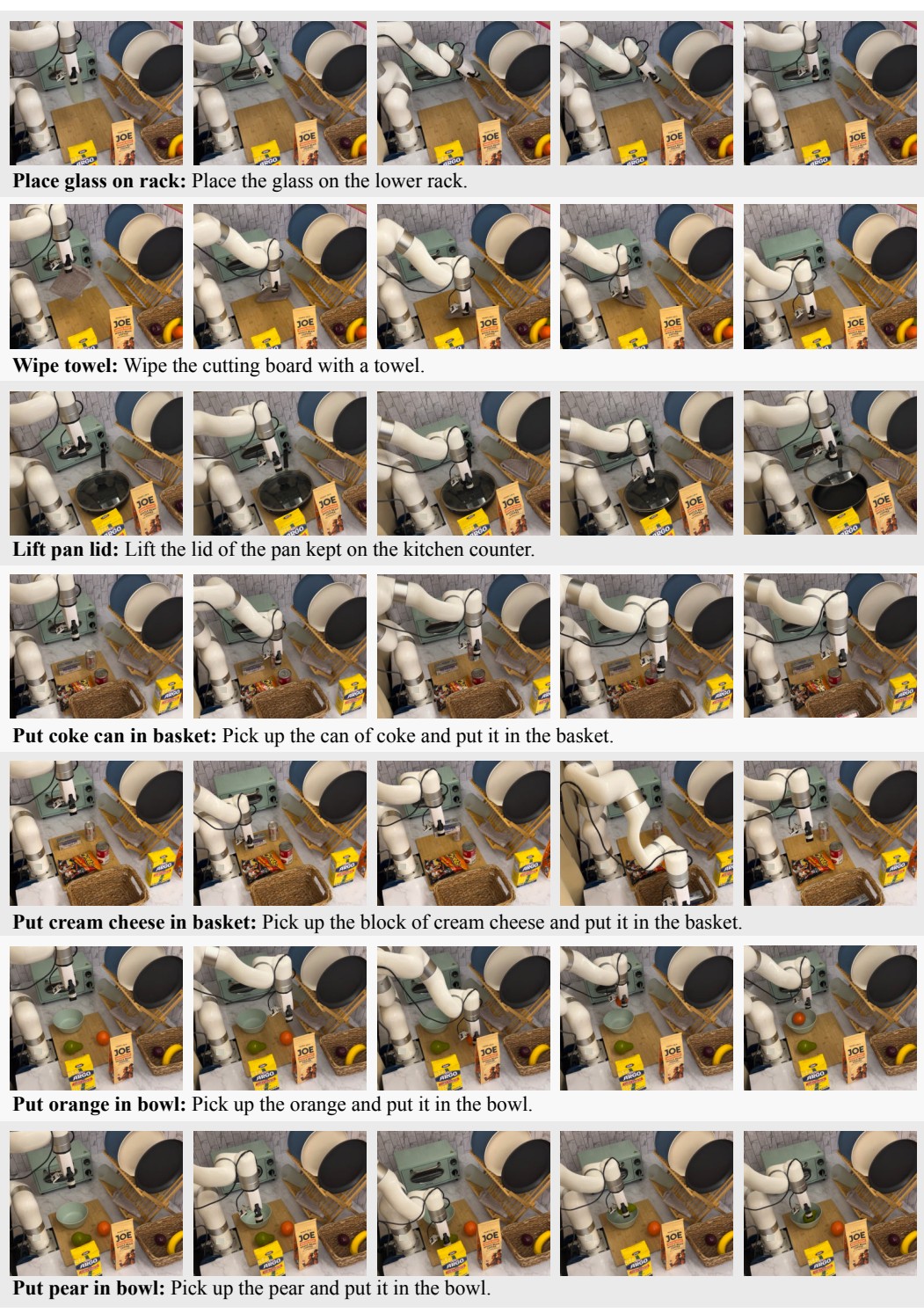

Figure 7: Real-world policy rollouts showing BAKU's capability in complex manipulation tasks.

with different multimodal heads. We observe that BAKU with different action heads is robust to the value of the random seed. Due to limited compute and the large number of multi-task experiments, we provide these results on the LIBERO-90 and Metaworld benchmarks.

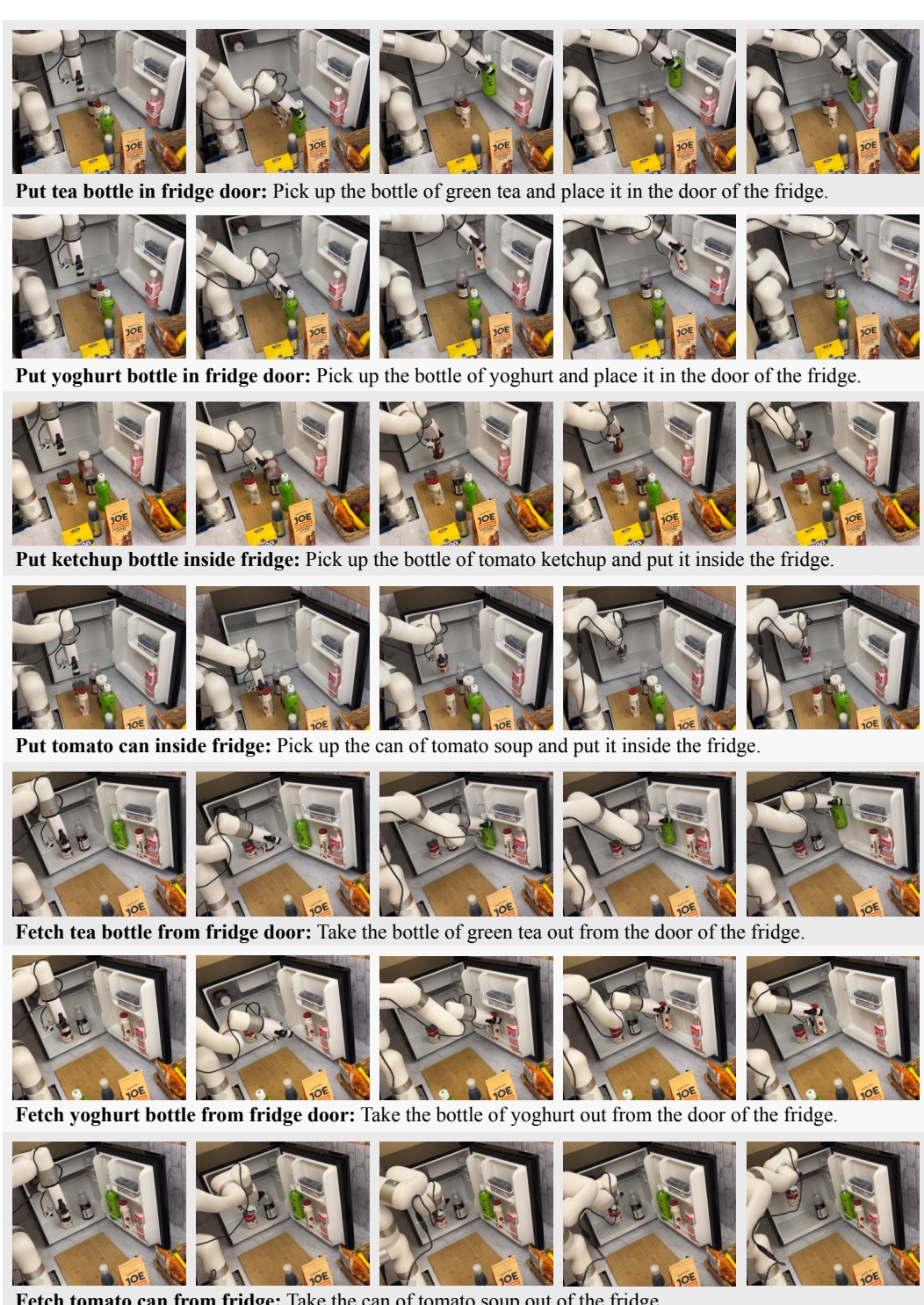

Figure 8: Real-world policy rollouts showing BAKU's capability in complex manipulation tasks.

**Observation trunk input** In our proposed architecture (see Section 3.4), the encoded observations from different modalities are passed individually as tokens into the observation trunk along with the action token to output the action feature representation. An alternative approach is to concatenate all the encoded inputs into a single vector and pass it through the observation trunk. As shown in Table 12, for Meta-World and DMC, which each have only a single input source, there is no

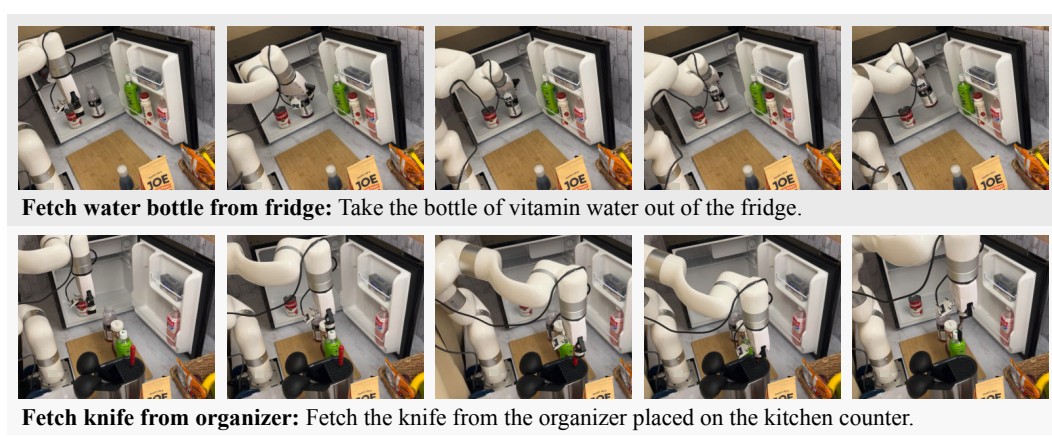

**Fetch water bottle from fridge:** Take the bottle of vitamin water out of the fridge.

**Fetch knife from organizer:** Fetch the knife from the organizer placed on the kitchen counter.

Figure 9: Real-world policy rollouts showing BAKU's capability in complex manipulation tasks.

**Set up table:** Place the white plate and the glass on the kitchen counter.

**Pick broom and sweep:** Pick up the broom and sweep the cutting board.

**Pick towel and wipe:** Pick up the towel from the lower rack and wipe the cutting board.

**Take bowl out of the oven:** Take the bowl out of the oven and place it on the kitchen counter.

**Put yoghurt inside and take water bottle out of fridge:** Put the yoghurt in the door of the fridge and take out the bottle of water from inside the fridge.

Figure 10: Real-world policy rollouts showing BAKU's capability on long-horizon manipulation tasks.

Table 6: Real task-wise performance

| Task | Number of Demonstrations | Successes (out of 5) | | | |
|---|---|---|---|---|---|
| | | RT-1 | MTACT | Baku | Baku w/ VQ-BeT |
| Fetch glass from rack | 20 | 5 | 5 | 5 | 5 |
| Fetch towel from rack | 28 | 5 | 2 | 5 | 5 |
| Fetch tea bottle from rack | 16 | 0 | 3 | 5 | 5 |
| Fetch water bottle from rack | 16 | 0 | 0 | 5 | 5 |
| Pick blue mug | 16 | 5 | 5 | 5 | 5 |
| Pick light blue bowl | 25 | 5 | 5 | 5 | 5 |
| Pick orange from bowl | 27 | 0 | 0 | 3 | 4 |
| Pick coffee bag | 19 | 3 | 5 | 5 | 5 |
| Pick box of corn starch | 14 | 0 | 3 | 5 | 5 |
| Lift blue plate from the rack | 18 | 0 | 4 | 5 | 5 |
| Lift white plate from the rack | 18 | 5 | 5 | 5 | 5 |
| Lift black plate from the rack | 12 | 2 | 3 | 5 | 5 |
| Open oven door | 17 | 0 | 0 | 0 | 3 |
| Close oven door | 27 | 0 | 3 | 3 | 4 |
| Place glass on rack | 19 | 5 | 5 | 5 | 5 |
| Wipe towel | 17 | 4 | 5 | 5 | 5 |
| Lift pan lid | 18 | 1 | 2 | 4 | 4 |
| Put coke can in basket | 19 | 0 | 0 | 3 | 3 |
| Put cream cheese in basket | 19 | 0 | 3 | 5 | 5 |
| Put orange into bowl | 14 | 0 | 0 | 4 | 5 |
| Put pear into bowl | 17 | 0 | 0 | 3 | 5 |
| Put tea bottle in fridge door | 18 | 0 | 0 | 1 | 0 |
| Put yoghurt bottle in fridge door | 17 | 3 | 5 | 3 | 5 |
| Put ketchup bottle inside fridge | 15 | 5 | 4 | 5 | 5 |
| Put tomato can inside fridge | 11 | 0 | 0 | 5 | 4 |
| Fetch tea bottle from fridge door | 11 | 5 | 5 | 5 | 5 |
| Fetch tomato can from fridge door | 11 | 0 | 1 | 5 | 5 |
| Fetch yoghurt bottle from fridge door | 10 | 0 | 3 | 5 | 4 |
| Fetch water bottle from fridge | 11 | 2 | 3 | 5 | 5 |
| Fetch knife from organizer | 20 | 0 | 5 | 5 | 5 |
| Mean | 17 | 1.83 | 2.8 | 4.3 | **4.53** |
| Mean success rate (out of 1) | – | 0.37 | 0.56 | 0.86 | **0.91** |

difference in performance, as expected. However, for LIBERO-90, which uses two camera views and the robot's proprioceptive state as inputs, there is a 3% absolute improvement in performance when using separate observation tokens as compare to a single concatenated vector.

# E  Broader Impacts

In this work, we present BAKU, a simple and efficient transformer architecture for multi-task policy learning. This work takes an important step toward enabling more efficient training of generalist robotic agents capable of performing diverse tasks, reducing the need for large datasets of expert demonstrations which are costly and time-consuming to collect. Further, BAKU focuses on improving data efficiency by maximally leveraging available training data, which is particularly valuable in robotics where data collection is expensive.

Table 7: Real task-wise performance for long-horizon tasks

| Task | Number of Demonstrations | Successes (out of 5) | |
|---|---|---|---|
| | | MTACT | Baku |
| Set up table | 34 | 3 | 3 |
| Pick broom and sweep | 13 | 4 | 5 |
| Pick towel and wipe | 14 | 2 | 4 |
| Take bowl out of the oven | 18 | 5 | 5 |
| Put yoghurt inside and take water bottle out of fridge | 17 | 2 | 4 |
| Mean | 19 | 3.2 | **4.2** |
| Mean success rate (out of 1) | – | 0.64 | **0.84** |

Table 8: Data efficiency analysis on the LIBERO-90 benchmark.

| # Demos | RT-1 | MT-ACT | BAKU |
|---|---|---|---|
| 5 | 0 | 0.31 | **0.58** |
| 10 | 0.01 | 0.48 | **0.71** |
| 25 | 0.04 | 0.49 | **0.83** |
| 50 | 0.16 | 0.54 | **0.9** |

Table 9: Data efficiency analysis on the Meta-World benchmark.

| # Demos | RT-1 | MT-ACT | BAKU |
|---|---|---|---|
| 5 | 0.40 | 0.07 | **0.59** |
| 10 | 0.49 | 0.10 | **0.67** |
| 25 | 0.62 | 0.11 | **0.76** |
| 35 | 0.65 | 0.13 | **0.79** |

Table 10: Performance of multi-task policies learned using BAKU on LIBERO-90 and Meta-World. We report the mean and standard deviation for each variant across 3 seeds.

| Method | LIBERO-90 (90 tasks) | Meta-World (30 tasks) |
|---|---|---|
| RT-1 | $0.14 \pm 0.02$ | $0.64 \pm 0.01$ |
| MTACT | $0.55 \pm 0.01$ | $0.12 \pm 0.01$ |
| BAKU (**Ours**) | $\mathbf{0.89 \pm 0.01}$ | $\mathbf{0.81 \pm 0.02}$ |

Table 11: Performance of BAKU with different action heads on LIBERO-90 and Meta-World. We report the mean and standard deviation for each variant across 3 seeds.

| Action Head | LIBERO-90 | Meta-World |
|---|---|---|
| MLP | $0.89 \pm 0.01$ | $\mathbf{0.81 \pm 0.02}$ |
| GMM | $0.83 \pm 0.02$ | $0.64 \pm 0.02$ |
| BeT | $0.88 \pm 0.01$ | $0.77 \pm 0.01$ |
| VQ-BeT | $\mathbf{0.9 \pm 0.01}$ | $0.78 \pm 0.005$ |
| Diffusion | $0.88 \pm 0.01$ | $0.64 \pm 0.01$ |

Table 12: Study of design decisions for the model architecture that affects multi-task performance.

| Category | Variant | LIBERO-90 | Meta-World | DMC |
|---|---|---|---|---|
| Separate vs. Shared Vision Encoders | Common | 0.90 | – | – |
| | Separate | **0.92** | – | – |
| Observation Trunk Input | Separate | **0.90** | **0.79** | **0.70** |
| | Concatenated | 0.87 | **0.79** | **0.70** |

