# OpenReview forum: "BAKU: An Efficient Transformer for Multi-Task Policy Learning"
_NeurIPS.cc/2024/Conference — NeurIPS 2024 poster_

### Official Review · Reviewer_Y1eK · 2024-07-08

**Soundness:** 3
**Presentation:** 4
**Contribution:** 3
**Rating:** 7
**Confidence:** 4

**Summary:**

The paper combines a series of architectural modifications made by prior work in offline imitation learning and shows that the new, composite architecture performs well in multi-task simulated settings and in data scarce, real-world robotics scenarios. Comprehensive ablations show the impact of each design choice made in selecting the final architecture.

**Strengths:**

- Strong results relative to the chosen baselines and in both simulated and real-world collections of tasks.
- Thorough investigation of different viable architecture designs.
- Comprehensive ablations.
- Excellent written communication.

**Weaknesses:**

- No variance or confidence intervals reported for results on simulated environments.
- Limited number of baselines, with RT-1 being somewhat outdated.
- It would be useful to see how performance varies across data scales for the different architectures (incl. ablations and baselines).

**Questions:**

- Why have you only considered two offline imitation learning baselines?
- Why do you not report variance in your results?

**Limitations:**

The paper appropriately addresses its limitations.

---

> ### Author Rebuttal · Authors · 2024-08-06
>
> We thank you for your positive response to our paper and we are glad that you like the paper writing and the results. We provide clarifications to your questions below.
>
>
> **“No variance or confidence intervals reported”:** We have included the results for multiple seeds with standard deviation in Table 1 and Table 2 of the PDF attached to the global rebuttal. We will update these results in the final version of the paper.
>
>
> **“Limited number of baselines”:** We would like to provide some clarification here. Our 2 primary baselines, RT-1 and MT-ACT, were selected as they propose different transformer-based architectures for offline imitation learning. Other popular methods include modifying the learning objectives to allow better modeling of the training data. These include GMMs, Behavior Transformers, and Diffusion Policy. We include the comparisons with these objectives in Table 2 showing that BAKU is a policy architecture that can incorporate the latest advances in policy learning objectives while learning from a smaller number of demonstrations and with a small model size.
>
>
> **“... how performance varies across data scales …”:** We thank you for bringing up this point. We have included a data efficiency analysis in the global rebuttal with the results provided in Table 4 and Table 5 of the attached PDF. We observe that at each level of data availability, BAKU shows a significantly higher success rate than MT-ACT and RT-1.
>
>
> We hope we have addressed your questions in the above clarifications. Kindly let us know if you have additional questions and we would be more than happy to discuss further.

---

> > ### Comment · Reviewer_Y1eK · 2024-08-08
> >
> > I thank the authors for their further engagement, clarifications and for including compelling results across different data scales and with different action heads. I maintain my score, as I see this as technically solid work that is well presented and demonstrates positive results.

---

> > > ### Author Response · Authors · 2024-08-11
> > > **Thank you**
> > >
> > > Thank you very much for your time and consideration.

---

### Official Review · Reviewer_RQGT · 2024-07-10

**Soundness:** 3
**Presentation:** 3
**Contribution:** 2
**Rating:** 6
**Confidence:** 4

**Summary:**

This paper presents a model for multi-task behavior cloning.  The paper analysis several different existing architectural options across three simulated environments to find the best options.  The best model is then tested in the real world across a series of tasks on real robots.

**Strengths:**

The paper tests on a large number of simulated and real world tasks.

The paper is very clear and understandable.

**Weaknesses:**

Most of the conclusions about architectural choices are based on experiments in simulation.  However, the real world results show that the ordering in the simulation doesn't always stay the same when the architecture is used in the real world.  Since we want these systems to work in the real world, it would be good to have more extensive experiments about when the ordering between architectures remains the same in the real world.  This is also important for when the simulation experiments had inconclusive results, or when minimal performance improvements in simulation were sacrificed for increased inference speed.

The paper doesn't include any methodological innovations, instead combining existing approaches.  While this is ok, it makes having a strong evaluation even more critical.

**Questions:**

Is RT-1 fine-tuned on the tasks, trained from scratch, or are the numbers its zero-shot performance?

Are the models trained using relative or absolute actions?  In my understanding, MLP action heads tend to work best with relative actions, while diffusion policies work best with absolute actions.  It is important to compare the various models under the most favorable conditions for each one, so it would be good to include experiments with and without absolute actions for each action head.

There are a few instances where choices that decrease success rate were chosen because of inference speed or model parameter count, such as choosing not to have shared vision encoders.  It would be good to have some numbers for what the inference speed tradeoffs were because other users of the model may want to make different choices.

**Limitations:**

The paper discusses its limitations.

---

> ### Author Rebuttal · Authors · 2024-08-06
>
> We thank you for your positive response to our paper and we are glad that you found the paper clear and understandable. We provide clarifications to your questions below.
>
>
> **“... more extensive experiments … real world”:** We include a new experiment studying the performance of BAKU on long-horizon tasks in the simulated LIBERO-10 benchmark and our real-world multi-task kitchen environment. We have included this in the global rebuttal. To summarize, we observe a similar trend where BAKU significantly outperforms our strongest baseline, MT-ACT, on these long-horizon tasks, achieving on average 19% higher success rate. We would also like to refer to line 171 in the paper where we mention that we perform 150 real-world evaluation trials per method. Doing this for the 4 methods in Table 1 brings the total count to 600 real-world trials which is very time-consuming. Hence, we restrict ourselves to using the simulated benchmarks for our ablation study. Moreover, from Table 1, LIBERO-90 is the most similar simulated benchmark to our real-world setup, with both using multiple camera views and robot proprioception as input. In this identical setting, we observe that the performance order of the baselines and BAKU remains the same with BAKU outperforming the baselines by a significant margin. We thank you for bringing up this important point and will include this explanation in the final version of the paper for more clarity.
>
>
> **“... doesn't include any methodological innovations, instead combining existing approaches”:** You are correct that we try to find the right design choices for multi-task policy learning in BAKU. We have addressed this concern in the global rebuttal. To summarize, we observe that small design choices in the architecture lead to massive improvements in multi-task policy performance. To the best of our knowledge, no prior work has looked into this. Appendix D further elaborates on the differences between BAKU and state-of-the-art prior work.
>
>
> **“Is RT-1 fine-tuned on the tasks, trained from scratch, or are the numbers its zero-shot performance?”:** All baselines in the paper and BAKU are trained from scratch and evaluated on the tasks they are trained on with randomized object initialization during evaluations. We primarily focus on the effect of architecture design choices on policy performance in this work and agree that understanding the zero-shot performance of multi-task policies is an interesting future direction.
>
>
> **“Are the models trained using relative or absolute actions?”:** The policies on LIBERO, Metaworld, and the real-world setup are trained on relative actions. We stick to the base configuration of the benchmarks that we evaluate. We will include this detail in the final version of the paper.
>
>
> **“MLP action heads … relative actions … diffusion policies … absolute actions”:** Thank you for this suggestion. Recent work [1] shows that even for diffusion policy, delta or relative actions show a better performance than absolute actions. We will be happy to add a comparison across different action spaces in the final version of the paper.
>
>
> **“... good to have some numbers for what the inference speed tradeoffs …”:** Thank you for the feedback. We agree that providing performance metrics is important for potential users. Accordingly, we have evaluated the inference time tradeoffs between using a shared vision encoder and separate view-specific encoders.  We tested scenarios with 2 camera views, matching the setup in LIBERO, and 4 camera views like the real-world xArm environment. A shared encoder allows batching frames from all views into a single forward pass, while separate encoders require individual passes.  The table below shows inference times for a single forward pass with BAKU:
>
>
> | Architecture | 2 Views | 4 Views |
> |-|-|-|
> | Shared Encoder| 9.14ms | 9.5ms |
> | Separate Encoders | 10.69ms | 14.91ms |
>
>
> We see a modest 3.94% increase when going from 2 to 4 views with a shared encoder, but a much larger 39% increase with separate encoders.  Further, our visual encoders have 1.4M parameters in a 10M parameter model. Separate encoders would result in 11.4M and 14.2M parameter models (for 2 and 4 camera views respectively), increasing the size by 14% and 42% respectively. We thank you for suggesting this study and we will include this in the final version of the paper.
>
>
> We hope we have been able to address your questions in the above clarifications. Kindly let us know if you have additional questions or concerns that stand between us and a higher recommendation for acceptance.
>
>
> [1] Chi, Cheng, et al. "Universal manipulation interface: In-the-wild robot teaching without in-the-wild robots." arXiv preprint arXiv:2402.10329 (2024).

---

### Official Review · Reviewer_FcXE · 2024-07-12

**Soundness:** 2
**Presentation:** 4
**Contribution:** 2
**Rating:** 5
**Confidence:** 4

**Summary:**

This work comprehensively investigates the architecture for multi-task policy learning on robotics manipulation tasks. Considering imitation learning with a limited number of demonstrations, the authors study the design choice of the network architecture, including encoders for observation with multi-modality, observations chunk, and action head. This work presents experiments on diverse benchmarks in the simulator and real-world robot manipulation tasks. According to the detailed ablation study of design choices on the components, this work selects the architecture design with a performance around 20% improvement over the baselines.

**Strengths:**

The presentation is clear and easy to understand

The authors conducted extensive experiments and solid evidence to support their proposed architecture

It is impressive to achieve >90% real-world performance with a limited number of demonstrations on the real robot.

**Weaknesses:**

The contribution of this work is mostly the empirical study of the combination of existing architecture/design choices. It does not propose any new architecture/design choices

Since the main contribution is empirical study rather than a new idea, it is important to share the code so that other researchers can replicate the experiments and benefit from this work for experiments on other tasks. However, the code is not provided along with this submission.

This work did not consider the time efficiency and data efficiency for multi-task policy learning when comparing different architectures and baselines. But these factors are also very important for robot learning. It will be much better to also discuss these factors.

**Questions:**

I’m concerned about the baseline implementation, especially RVT-1 on LIBERO and real-robot tasks, since it seems much poorer than RVT-1 paper (though RVT-1 shows experiments on RLBench and a different set of real-robot tasks.)

About keyframe extraction, which is an important part of the success of RVT-1, how do you define keyframes on LIBERO and real-robot data?

About action prediction, did you use the same discretization way as presented in RVT-1 paper?

Is it possible to study BAKU on RLBench, so we can have a more convincing comparison?

Do you have any intuitive explanation about why the baseline fails? RVT-1 also works well on real-world tasks in their paper. Their task set also consists of general pick&place tasks. Why does its success rate drop a lot in the task set in this submission? Only the architecture difference from BAKU contributes to the performance gap?

This work will be more convincing if the baselines can be discussed more in detail.

**Limitations:**

This work studies the architecture design with detailed ablative study, which is definitely informative and useful for other researchers in the area. However, it does not propose any new ideas/insights. So it is more like a technical report or system engineering paper, rather than a research paper.

Its contribution only focuses on the specific area of robotics manipulation, and the evidence or conclusions seem not generalizable/beneficial for any other AI area. Not sure whether NeurIPS is the proper venue. Perhaps it is more appropriate to be presented in a robotics conference/journal.

---

> ### Author Rebuttal · Authors · 2024-08-06
>
> We thank you for your insightful comments about the paper and we are glad that you found the paper easy to understand and the experiments extensive. We provide clarifications to your questions below.
>
>
> **“... combination of existing architecture/design choices”:** You are correct that we try to find the right design choices for multi-task policy learning in BAKU. We have addressed this concern in the global rebuttal. To summarize, we observe that small design choices in the architecture lead to massive improvements in multi-task policy performance. To the best of our knowledge, no prior work has looked into this. We have further elaborated the differences between BAKU and state-of-the-art prior work in Appendix D.
>
>
> **“... code is not provided along with this submission”:** Thank you for pointing this out. We completely agree that making the code public is essential for such a project and have added an anonymized version of the code to our anonymous webpage. Following the updated instructions, we have also privately shared the code with the AC.
>
>
> **“... did not consider the time efficiency and data efficiency for multi-task policy learning …”:** We agree with the reviewer that these are very important factors for robot learning, especially when learning tasks on a real robot. Accordingly, we have included a data efficiency study in the global rebuttal and would refer the reviewer to Table 4 and Table 5 in the PDF attached to the global rebuttal. To summarize, we observe that at each level of data availability, BAKU shows a significantly higher success rate than MT-ACT and RT-1.
>
>
> **“RVT-1 on LIBERO:”** We believe that there has been a confusion here. We compare BAKU with RT-1 which is another policy architecture that uses transformers. As you mentioned, RVT assumes the possibility of keyframe extraction so we do not compare BAKU against this line of work.
>
>
> **“...  only focuses on the specific area of robotics manipulation”:** In addition to robotic manipulation tasks in the LIBERO and Metaworld benchmarks and our real-world experiments, we have also included locomotion tasks in the DeepMind Control suite (DMC) in Table 1 and Table 2 to highlight the usefulness of the proposed architecture in domains other than manipulation.
>
>
> We hope we have been able to address your questions in the above clarifications. Kindly let us know if you have additional questions or concerns that stand between us and a higher score.

---

> ### Comment · Reviewer_FcXE · 2024-08-11
> **Thanks for Author Response**
>
> Thanks for the authors' response. I appreciate the code sharing and the detailed clarification. But I'm still concerned about the novelty of this work. Overall, I slightly increase my score.

---

> > ### Author Response · Authors · 2024-08-11
> > **Thank you**
> >
> > Thank you for raising the score from 4 to 5. We appreciate all the comments and suggestions.

---

### Official Review · Reviewer_eFXw · 2024-07-13

**Soundness:** 3
**Presentation:** 3
**Contribution:** 2
**Rating:** 5
**Confidence:** 4

**Summary:**

The paper presents a novel transformer architecture, BAKU, designed for efficient multi-task policy learning in robotics. BAKU consists of three components: sensory encoders, an observation trunk, and an action head. According to the experimental results, BAKU is showcased to be effective in both simulated and real-world tasks, offering a promising solution to the challenge of training generalist policies for physical robots.

**Strengths:**

1. The paper is well organized, and the writing is clear and well-understood. The author clearly describes the algorithm BAKU, including the objective of the optimization process and the details of the algorithm implementation.
2. The authors conduct extensive experiments compared to other baseline methods in both simulated and real-world tasks, as well as detailed ablation studies. The results show that BAKU outperforms other baselines by a large margin and can accomplish tasks using few demonstrations with high data utilization and efficiency.

**Weaknesses:**

1. The novelty of the algorithm presented in this paper is limited, as all components of the BAKU model are derived from other papers. For instance, RT-1 employs a transformer architecture and FiLM layers, while ACT utilizes action chunking techniques. The authors should discuss in greater detail how this algorithm differs from these previous works.
2. As indicated in the paper's title, "Efficient Transformer," it is essential to clarify what makes it efficient. Does efficiency pertain to the model's size, or is it the high success rate achieved with a small number of demonstrations? Additionally, which component is critical for enhancing efficiency?
3. Even though the experimental results indicate that this algorithm significantly outperforms RT-1 and MTACT, it remains unclear why it achieves such improvements. From a design perspective, BAKU does not propose a novel network architecture or optimization objective compared to RT-1 and MTACT; it rather appears to integrate elements from both.
4. In Table 2, I noticed an intriguing phenomenon: the 10M model performs the best, even surpassing the 114M model. What could be the reason behind this outcome?

**Questions:**

Details are listed above.

**Limitations:**

YES

---

> ### Author Rebuttal · Authors · 2024-08-06
>
> We thank you for your comments about the paper and we are glad that you found the paper well organized and the experiments extensive. We provide clarifications to your questions below.
>
>
> **“... novelty of the algorithm … limited”:** We have addressed the novelty of the algorithm in the global response. To summarize, we observe that small design choices in the architecture lead to massive improvements in multi-task policy performance. To the best of our knowledge, no prior work has looked into these design decisions. We have further elaborated the differences between BAKU and state-of-the-art prior work in Appendix D.
>
>
> **“... it is essential to clarify what makes it efficient”:** Thank you for pointing this out. The efficiency in this work pertains to both your points - a small model size (resulting in high frequency execution on a real robot) and a high success rate achieved with a small number of demonstrations (also evident from Table 4 and Table 5 in the PDF attached to global rebuttal). The explanation provided in Appendix D compares BAKU with current state-of-the-art policy architectures such as MT-ACT and RT-1 and highlights the source of this performance improvement. Given the importance of this concern, we will move Appendix D to the main paper in the final version to address any concerns about novelty.
>
>
> **“... unclear why it achieves such improvements”:** We would again like to refer you to Appendix D which highlights the differences between BAKU and RT-1/MT-ACT. You are correct that BAKU integrates useful ideas from several prior works and we show, through an extensive ablation study (Table 2), the effect of each of these components on the policy performance.
>
>
> **“... 10M model performs the best, even surpassing the 114M model”:** We have included an explanation for this result in line 199 (Section 4.3). In recent years, we have witnessed that larger models tend to require more data [1, 2]. However, when it comes to robotics, collecting large amounts of data can be challenging. For instance, the LIBERO benchmark only has 50 demonstrations available per task. We suspect that operating in this low data regime leads the 114M parameter model to overfit the training data, resulting in poor performance.
>
>
> We hope we have been able to address your questions in the above clarifications. Kindly let us know if you have additional questions or concerns that stand between us and a higher score.
>
>
> [1] Hoffmann, Jordan, et al. "Training compute-optimal large language models." Proceedings of the 36th International Conference on Neural Information Processing Systems. 2022.
>
> [2] Hernandez, Danny, et al. "Scaling laws for transfer." arXiv preprint arXiv:2102.01293 (2021).

---

> > ### Author Response · Authors · 2024-08-12
> >
> > Hi,
> >
> > With the discussion period ending soon, we wanted to ask if you have any remaining questions or concerns about our work. We would be happy to answer any further questions that you might have.

---

> > ### Comment · Reviewer_eFXw · 2024-08-13
> >
> > Thank you for the detailed response. Your rebuttal has addressed my concerns, so I would like to raise my final rating.

---

> > > ### Author Response · Authors · 2024-08-13
> > > **Thank you**
> > >
> > > Thank you for raising the score from 4 to 5. We appreciate all the comments and suggestions.

---

### Official Review · Reviewer_76DH · 2024-07-27

**Soundness:** 2
**Presentation:** 3
**Contribution:** 2
**Rating:** 6
**Confidence:** 5

**Summary:**

This paper presents a multitask BC model for robot learning. They systematically compare several components of recently proposed BC models and combines their best aspects into a common architecture. Key design choices include: observation trunk, model sizes, action heads, goal modalities, action chunking, observation history etc. Experimental comparison with RT-1, MT-ACT and BAKU-variants on 3 simulated benchmarks and on 30 real world tasks validates their choices and design methodology.

**Strengths:**

This paper presents a very systematic ablation study across various model components proposed in prior works. The final model is simple, it debunks some overcomplicated/compute-expensive choices like GMM/diffusion heads, observation history or FiLM etc. and highlights importance of others like Transformer trunk, action chunking etc.

**Weaknesses:**

1. The statistical significance of the experiments is questionable. For probabilistic approaches (GMM/diffusion) especially, a single unfavorable random seed could significantly skew the performance results for a baseline run.
2. Both LIBERO and MetaWorld environments offer 50 different initial states for each task, yet the reported results are based on only 10 rollouts. Many methods are known to be sensitive to out-of-distribution (OOD) initial states, but the paper lacks details on how initial states were selected for both LIBERO and MetaWorld experiments. This omission severely undermines the reliability and generalizability of the presented results. Furthermore, for such low number of rollouts, it becomes crucial to conduct evaluations across multiple random seeds to ensure the robustness of the findings.
3. The results reported for MT-ACT on MetaWorld appear to be notably lower than those presented in other recent works, such as [PRISE] (fig:15|80%) and [QueST] (fig:3|90%). This discrepancy raises questions about the MT-ACT implementation and evaluation in this work.
4. This work does not offer many new insights to the community. For example usefulness of probabilistic action heads over deterministic ones for multimodal data have been shown in [Robomimic] paper. Action chunking helps in libero, but the chunk size is a more sensitive parameter which has not been studied. Transformers over MLP and model size overfitting is a common knowledge.
5. The proposed model is very similar to BC-ResNet-Transformer from [Libero] paper, but still performs much better. This is not addressed in the paper.
6. Line 45 claims randomized object initialization but line 170 mentions initialization from fixed set during evaluation. This claim should be removed from main contributions if later is the case.

[PRISE]: https://arxiv.org/abs/2402.10450
[QueST]: https://arxiv.org/abs/2407.15840v2
[Libero]: https://arxiv.org/abs/2306.03310

**Questions:**

1. How can reported results be considered reliable given single training seed and limited evaluation rollouts?
2. Could you describe the initial state selection process for LIBERO and MetaWorld experiments?
3. Why are MT-ACT MetaWorld results lower compared to those reported in other aforementioned works?
4. What key differences between BAKU and BC-ResNet-Transformer from LIBERO contribute to BAKU's superior performance?

**Limitations:**

Limitations are adequately addressed.

---

> ### Author Rebuttal · Authors · 2024-08-06
>
> We thank the reviewer for the detailed comments about the work. We are glad that you found the study systematic and would like to provide clarifications to the questions below:
>
>
> **“The statistical significance of the experiments is questionable”:** We have conducted additional experiments and report the mean and standard deviation across 3 different seeds for the simulated environments in Table 1 and Table 2 of the PDF attached to the global rebuttal. We find the performance of the model is quite robust to seeds and hence significant.
>
>
> **Initial state selection for LIBERO and Metaworld:** For both simulated benchmarks, we reset the environment to a random state and evaluate on the sampled state. We have done this across 3 different seeds for each method to avoid a favorable seed value from impacting the results.
>
>
> **Lower performance of MT-ACT compared to other works:** Thank you for bringing these works to our notice. We believe there might be some confusion here. Both PRISE and QueST use 100 demonstrations for each task on Metaworld as compared to the 35 demonstrations per task used by us. Further, both these papers also run MT-ACT on LIBERO-90 using the same 50 demonstrations per task provided by the benchmark. The multitask MT-ACT performance on LIBERO-90 reported by PRISE is 46.6% (Figure 6) and by QueST is also 46.6% (Figure 2a). Under the same setting, we report a multitask MT-ACT performance of 54% (Table 1). Hence, we believe that the small dataset size used in BAKU is the cause of lower performance in Metaworld.
>
>
> **“... key differences between BAKU and BC-ResNet-Transformer from LIBERO …”:** We would like to note that the LIBERO paper primarily focuses on lifelong learning and provides a comparison with multi-task learning performance (48% success in Table 2 and Figure 5 of the paper). Though we primarily evaluate BAKU in the multi-task learning setting, we want to point out that BAKU is also compatible with lifelong learning. There are some key differences between BAKU and the BC-ResNet-Transformer (BC-ResNet-T) architecture used in LIBERO. While BC-ResNet-T uses a Gaussian mixture model (GMM) for its action head, the vanilla version of BAKU uses a simpler deterministic head with action chunking, which significantly improves performance (Figure 1b and Table 2). Additionally, BAKU shows that state-of-the-art multimodal policy classes like VQ-BeT and Diffusion Policy can be unified under a common framework. Further, BC-ResNet-T incorporates a history of observations. We show in Table 2 that using history without a multi-step loss can hurt performance and could be a potential reason behind the low performance of BC-ResNet-T. Hence, BAKU, with its simple design choices, results in a significant performance boost on the same LIBERO tasks as compared to BC-ResNet-T.
>
>
> **“Line 45 claims randomized object initialization but line 170 mentions initialization from fixed set during evaluation”:** Thank you for pointing this out. We meant that the objects are initialized at random locations and we use the same set of randomized locations for all baselines to keep the comparisons fair. We will make this clearer in the final version of the paper.
>
> We hope we have been able to address your questions in the above clarifications. Kindly let us know if you have additional questions or concerns that stand between us and a higher score.

---

> > ### Comment · Reviewer_76DH · 2024-08-11
> >
> > I thank authors for reporting additional seeds, it definitely builds confidence into the results and hence the conclusions. Most of my other concerns have been addressed except a few as follows:
> >  - I re-implemented BC-ResNet-T from Libero and got 84.4% success rate on Libero-90, this is without action chunking, with a GMM head and observation history of 10. This number is quite close to BAKU with GMM head, thus suggesting action chunking and observation history does not affect as much (at least on Libero and Metaworld benchmarks).
> >  - What matters more is the action head and as per results a simple MLP head does much better. This is contrary to the findings of Robomimic paper, where no-GMM head led to perf drop and their conclusion that GMM heads help with capturing multimodality in human demonstrations. I would like to know authors take on this, as well on if BAKU with MLP head might struggle with multimodal demonstrations?
> >
> > I still have my reservations about the novelty of proposed architecture, as observation trunk with action heads is a well explored idea for example in BC-Resnet-T (BAKU with GMM head) and OCTO (BAKU with diffusion head). However, I acknowledge and appreciate the systematic comparison that BAKU makes and the practical insights it offers, which are also validated in real-robot experiments. Considering this I have updated my score.

---

> > > ### Author Response · Authors · 2024-08-11
> > > **Thank you**
> > >
> > > Thank you for raising your score from 3 to 6. We truly appreciate your consideration and constructive feedback.
> > >
> > >
> > > To address your remaining concerns:
> > >
> > >
> > > **“... re-implemented BC-ResNet-T from Libero … 84.4% success rate …”:** This is interesting. We assume that you trained the policy with a multi-step loss resulting in a higher performance than reported in the LIBERO paper. Thank you for sharing this result.
> > >
> > >
> > > **“... if BAKU with MLP head might struggle with multimodal demonstrations?”:** Thank you for the question. The high performance of the MLP head in BAKU shows that a simple MLP can result in surprisingly high performance and is probably something we should try as a first step when training robot policies. That being said, we do believe that multimodal heads would work better than modeling the action space as an unimodal Gaussian when the data is multimodal. This is also corroborated by our real-world experiments where BAKU with a VQ-BeT head performs better than with an MLP head.
> > >
> > >
> > > We thank you again for your consideration and we will be happy to address any further questions or concerns that you might have.

---

### Author Rebuttal · Authors · 2024-08-06

We thank the reviewers for their constructive feedback. The reviewers have requested important clarifications and additional experiments. We have provided these as a detailed response to each review along with a summary of some shared concerns in this global response.

**Concerns about novelty (Reviewers 76DH, eFXw, FcXE, RQGT):** Multiple reviewers have raised concerns regarding the novelty in BAKU. We agree that our work is focused on finding the right architectural choices for multi-task policy learning. Importantly, we observe that simple design choices lead to massive improvements in policy performance, achieving a 36% improvement on the LIBERO-90 benchmark and a 35% improvement on our real-world tasks. We view this simplicity as a strength of our work that will allow the community to build on this by making minor changes to get good performance. We believe that a deeper understanding of the novelty of the architecture choices can be found in Appendix D which compares BAKU with current state-of-the-art policy architectures such as MT-A​​CT and RT-1 and highlights the architectural choices in BAKU that lead to significant performance improvement. Given the importance of this concern, we will move Appendix D to the main paper in the final version to address any concerns about novelty.

**Experiments including multiple seeds (Reviewers 76DH, Y1eK):** We provide results on BAKU, RT-1, and MT-ACT across 3 seeds in Table 1 in the attached PDF. We observe that all three methods are robust to different seed values. Reviewer 76DH suggested that probabilistic approaches like GMM and diffusion are sensitive to favorable seed values and evaluating on a single seed makes the result unreliable. Thus, Table 2 in the attached PDF includes results across 3 seeds on BAKU with different multimodal heads. We observe that BAKU with different action heads is robust to the value of the random seed. Due to limited compute and the large number of multi-task experiments, we provide these results on the LIBERO-90 and Metaworld benchmarks. More seeds and the DM Control environment will be added in the final version of the paper.

**New experiment on long-horizon tasks (Reviewer RQGT):** We include a new experiment studying the performance of BAKU on long-horizon tasks in the simulated LIBERO-10 benchmark and our real-world multi-task kitchen environment. Table 3 in the attached PDF provides the results on 10 tasks in LIBERO-10 and 5 tasks in the real kitchen environment, each composed of two shorter tasks chained sequentially. We use 50 demonstrations per task for LIBERO-10 and an average of 19 demonstrations per task for the real robot. We observe that BAKU significantly outperforms our strongest baseline, MT-ACT, on these long-horizon tasks, achieving on average 19% higher success rate. This highlights BAKU's ability to learn policies that can effectively plan and execute sequences of actions over extended time horizons. We include the real-world rollouts of these long-horizon tasks in Figure 1 of the attached PDF and have also included rollout videos on our anonymous webpage (link in the paper). We provide a description of the real-world tasks below for reference.

| Task                            | Description                                                       |
|---------------------------------|-------------------------------------------------------------------|
| Set up table                    | Place the white plate and the glass on the kitchen counter        |
| Pick broom and sweep            | Pick up the broom and sweep the cutting board                     |
| Pick towel and wipe             | Pick up the towel from the lower rack and wipe the cutting board  |
| Take bowl out of the oven       | Take the bowl out of the oven and place it on the kitchen counter |
| Yoghurt in, water out of fridge | Put yoghurt inside and take water bottle out of the fridge        |
| | |

**Data efficiency analysis (Reviewer FcXE):** We analyze the performance of BAKU with varying number of demonstrations in Table 4 and Table 5 of the attached PDF. We observe that at each level of data availability, BAKU shows a significantly higher success rate than MT-ACT and RT-1.

**Code availability (Reviewer FcXE):** We completely agree that making the code public is essential for such a project and have added an anonymized version of the code to our anonymous webpage (link in the paper). Following the updated NeurIPS instructions, we have also privately shared the code with the AC.

We have addressed other concerns specific to each reviewer in their rebuttal section. We hope that these updates to our paper inspire further confidence in our work. At the same time, we invite any further questions or feedback that you may have on our work.

---

### Decision · Program_Chairs · 2024-09-25

**Decision:**

Accept (poster)

**Comment:**

The paper presents a multitask Behavior Cloning (BC) model for robot learning, comparing various components of recent BC models and combining their best aspects.
Some of the concerns remains on the novelty of the algorithm presented in this paper is limited, as all components of the BAKU model are derived from other papers. However, the authors provide a very detailed set of experiments to
Key design choices include observation trunk, model sizes, action heads, goal modalities, and action chunking. Experiments on simulated benchmarks and real-world tasks validate their design. The study highlights the effectiveness of simpler components like Transformer trunks and action chunking, while questioning the need for more complex elements.

Furthermore, the statistical significance of the experiments is debatable due to limited rollouts and potential sensitivity to initial states. Furthermore some of the reviewers also pointed out time efficiency.  The rebuttal phase resulted in a number of clarifications, additional and subsequent score updates.
Overall the meta reviewer finds that there is a significant  technical value in summarizing the empirical ideas into best practices, while trimming the unnecessary algorithmic components. This papers does a good job at it and provides a strong experimental evaluation.   The authors are request to release clean and reusable code along with model checkpoints for reproducibility of results and evaluations.  Furthermore, the authors should also update the papers using feedback from the rebuttal phase and additional experiments.